

# Boat anchoring contributes substantially to coral reef degradation in the British Virgin Islands

Rebecca L. Flynn[*] and  Graham E. Forrester[*]

Department of Natural Resources Science, University of Rhode Island, Kingston, RI, United States of America
[*] These authors contributed equally to this work.

## ABSTRACT

Community decline is often linked to anthropogenic activities. Coral reef declines, for example, have been linked to overfishing and climate change, but impacts of coastal development, pollution, and tourism have received increasing attention. Here, we isolated the impact of one little-studied aspect of recreational activity on coral reefs—damage from boat anchoring—by performing a survey of 24 sites in the British Virgin Islands (BVI) subject to varying levels of anchoring activity. The percent cover of hard corals and sea fans was reduced by a factor of ~1.7 and ~2.6 respectively at highly anchored sites. Hard coral colonies were  40% smaller in surface area and ~60% less dense at sites experiencing high anchoring frequency. In addition, highly anchored sites supported only ~60% of the species richness of little anchored sites. Frequently anchored sites were ~60% as structurally complex and supported less than half as many fish as those rarely anchored, with 5 of 7 fish functional groups affected. Roughly 24% of BVI coral reef by area appears suitable for anchoring, suggesting that impacts associated with boat anchoring may be both locally severe and more pervasive than previously appreciated.

## INTRODUCTION

Human activities are degrading habitats on a global scale, resulting in a loss of biodiversity, trophic collapse, and diminished ecosystem function and services (*Ehrlich & Wilson, 1991*; *Naeem et al., 1994*; *Dobson et al., 2006*). Coral reefs, in particular, are high diversity habitats accounting for approximately one quarter of the ocean's biodiversity while occupying less than 0.01% of the marine environment (*Burke et al., 2011*). Reefs perform several ecosystem services by protecting shorelines, supplying fisheries, and attracting tourism and recreation that provide nations with revenue (*Burke et al., 2011*). Coral reefs are, however, declining globally (*Gardner et al., 2003*; *Schutte, Selig & Bruno, 2010*) and losing three-dimensional complexity (*Alvarez-Filip et al., 2009*). Both diminishing coral cover and complexity negatively impact reef fish, some of which rely on live coral for food while others utilize the structure as refuge (*Lewis, 1998*; *Graham et al., 2009*).

Corresponding author
Graham E. Forrester,
gforrester@uri.edu

Reef degradation is caused by the integrative effects of natural disturbances, such as hurricanes, and anthropogenic stressors (*Wilkinson & Buddemeier, 1994*); (*Wilkinson, 2008*). Key anthropogenic stressors include global climate change (ocean warming and acidification) and local effects from invasive species, overfishing, coastal pollution and other symptoms of increasing human population density (*Wilkinson, 2008*; *Jackson et al., 2014*). Boat anchoring is one symptom of increasing human visitation (*Jackson et al., 2014*) that may contribute to reef degradation. As ocean recreation and the associated boat traffic increase rapidly in many areas of the world (*Davenport & Davenport, 2006*; *Burgin & Hardiman, 2011*), physical damage to reefs may also increase. Physical damage from boat anchors and the associated chains (hereafter, collectively referred to as anchors or anchoring) is an acknowledged source of damage to coral reefs (*Goenaga, 1991*). Compared to other human impacts, however, boat anchoring has been the subject of virtually no formal study (*Johnstone, Muhando & Francis, 1998*). By way of illustration, a search of Web of Science for "coral reef and anchor" returned only 68 papers, whereas a search for "coral reef and climate" returned 2,335 and one for "coral reef and fishing" returned 6,234.

Previous studies have rarely addressed the community-wide impacts from *chronic* anchor damage. Our objective was, therefore, to quantify effects of chronic anchor damage to coral reefs through a replicated comparison of anchor-damaged and undamaged reefs. We assessed both the direct impacts on sessile species like corals and sea fans, and the indirect effects on reef-associated fishes. We expected that reefs that have experienced higher anchoring activity would support fewer and smaller corals. Consequently, we also expected anchor-damaged reefs would have lower structural complexity and lower densities of reef-associated fishes.

## MATERIALS AND METHODS

### Study location

We studied the effects of anchoring on reefs in the British Virgin Islands (BVI) because the number of active vessels and their size contribute to a high risk of anchor damage to reef habitats in this territory. Approximately 1100-1500 yachts (12–16 m in length) operate in the 155 km$^2$ of BVI water. This fleet is expanding and visitation by larger mega-yachts exceeding 45 m in length is increasing (personal communication with Janet Oliver, BVI Charter Yacht Society, and Trish Baily, BVI Association of Reef Keepers and BVI ReefCheck).

### Study design

We surveyed 24 reefs (each 0.5–0.75 ha in area) during the summer of 2014 to determine the effects of chronic anchoring. Sites were categorized as experiencing little or no anchoring (hereafter low, $n = 11$), occasional anchoring (hereafter medium, $n = 3$) or regular anchoring (hereafter high, $n = 10$) (Fig. 1). Site selection and classification was based primarily on the plausibility of use as an anchorage and expert opinion about the level of anchoring activity. We selected a set of candidate sites that were plausible anchorages because they were situated on the leeward sides of islands, usually near sand, and often

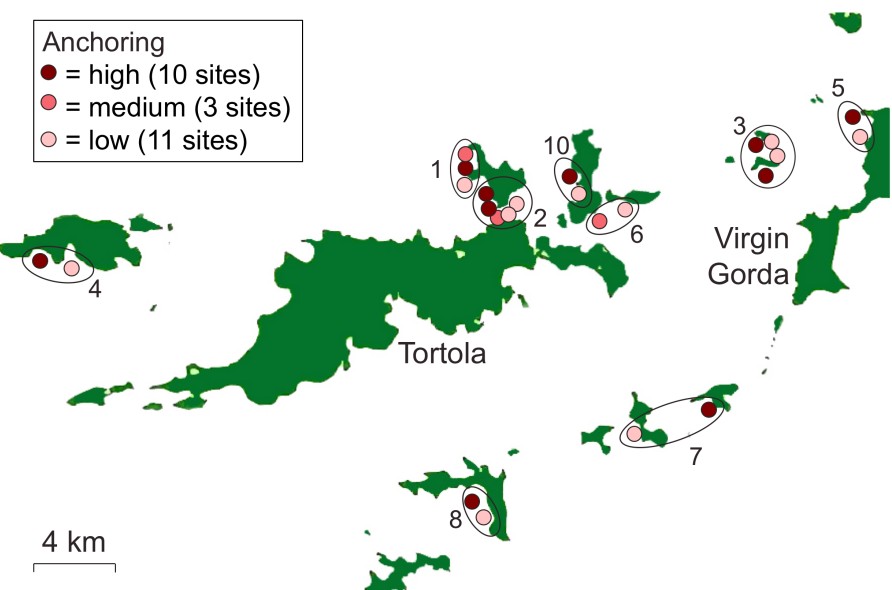

**Figure 1 Locations of reefs surveyed in the British Virgin Islands.** Sites are coded to indicate whether they categorized as experiencing high, medium, or low frequency of anchoring. Sites were selected in 10 groups (numbered and circled) that were close in proximity.

in bays. We also selected candidate sites we considered unlikely to be used as anchorages because, despite being on the leeward side of islands, they were locations likely to be unsafe (e.g., near a cliff) or undesirable (e.g., subject to heavy powerboat traffic) anchorages. We consulted seven local experts who all had >10 years of experience with boating activity in the BVI (dive instructors, charter captains, and marine conservation professionals) to classify sites. High- and low-anchoring sites were those for which all 7 experts agreed on the extent of anchoring. Medium-anchoring sites were those for which there were differences of opinion among local experts about the frequency of anchoring and, although they represent a less clearly-resolved category, we retained them for our surveys.

We performed two post-hoc checks to corroborate the classification of anchoring activity and to clarify the status of medium-anchoring sites. First, we quantified the observed density of yachts using satellite images of each site (Fig. S1). We used all available images from 2004–2014 that were not obscured by clouds ($n = 4$–10 images per site). There were boat moorings at some sites but, because we knew the locations of moorings, we could distinguish moored boats from those at anchor. Although the images provided a far smaller sample of observations than the collective observations of local experts, we predicted that mean yacht densities in the images would differ among low-, medium-, and high-anchoring sites. Secondly, during our SCUBA surveys at each site we quantified symptoms of anchor damage (described further below) and tested whether they occurred at different frequencies among low-, medium- and high-anchoring sites.

To control for other sources of variation among sites, we selected sites that were similar in depth, wave exposure, and reef slope. In addition, we selected low- and high-anchoring

sites in 10 groups that were close in proximity to further reduce the possibility that factors other than anchoring, particularly those associated with land-based human activity, differed among sites (Fig. 1). As a post-hoc check that we had not inadvertently created a bias among treatments by confounding anchoring with effects of land-based human activity, we used satellite images to measure the distance from the center of each reef site to the nearest shore and to the nearest developed land area (*Lirman & Fong, 2007*) (Fig. S2).

Finally, to determine what percentage of reef in the BVI is potentially vulnerable to anchoring we used GIS to classify coral reefs by exposure (leeward or windward), based on the assumption that only leeward reefs are potential anchoring sites. The GIS feature class utilized was a benthic habitat map showing areas of the sea floor covered by coral reef, seagrass, hard bottom, and algae (*NOAA et al., 2001*). We isolated the coral reef polygons, created a new geodatabase feature class of those areas, and edited the attribute table to include categories for leeward or windward exposure (*Environmental Systems Research Institute, ESRI, 2011*).

## Survey methods

We sampled each site on SCUBA using 3–8 haphazardly placed 30-m transects. We used the point-intercept method to estimate the percent cover of major benthic taxa, including live hard coral; sea fans; branching soft corals; fleshy, filamentous, calcareous and crustose algae; erect and encrusting sponges; and substrates, such as sand and rubble (*Almada-Villela et al., 2003*). In addition, all scleractinian coral colonies intersected by the transect tape were identified to species, classified by morphology, and measured in length (L) and maximum orthogonal width (W). Colony surface area (SA) was calculated assuming colonies were elliptical ($SA = 0.5L \times 0.5W \times \pi$). We calculated coral colony density using the Strong Method, in which density (organisms per m$^2$) $= \Sigma$ (1/M)(unit area/total transect length) where $M$ is the maximum orthogonal width (*Strong, 1966*; *Bakus, 2002*). We also calculated the number of coral species intercepted per transect, as a simple estimate of species richness that is adjusted for sampling effort. All individuals of the most common sea fan (*Gorgonia ventalina*) within a $30 \times 1$-m belt transect were counted and measured for height and width.

As a post-hoc check of our classification of anchoring activity at each site, we quantified symptoms consistent with recent anchor damage. For corals, we counted the number of colonies that were newly overturned, broken, or scarred with gouges that were consistent with being scraped by an anchor or chain. For sea fans, we counted the number of colonies that were recently bent or broken at the base, or contained large rips in colony tissue with no associated symptoms of disease.

To quantify the indirect effects of anchoring, we assessed reef complexity and fish population densities. The three-dimensional structural complexity (or rugosity) was estimated for each transect using the consecutive height difference method, for which the height of the tape off the bottom is measured every 50 cm and the square root of the sum of the squared differences between successive height measurements is used to describe vertical complexity (*McCormick, 1994*). Reef-associated fish were counted using $30 x 1.5$ m belt transects (45 m$^2$ in area). We counted all small- to medium-sized diurnal fish species, excluding very small cryptic benthic species (e.g., some gobies and blennies) and very mobile

mid-water species (e.g., jacks). Each fish species was classified by both size (juveniles = individuals judged to be <1 month post-settlement, usually <3 cm in total length; adults = all larger individuals) and trophically based functional group (macrocarnivores, piscivores, mobile benthic invertivores, sand invertivores, coral/colonial invertivores, spongivores, diurnal planktivores, nocturnal planktivores, territorial gardeners, turf grazers, scrapers, excavator/eroders, macroalgae grazers, general omnivores) (*Halpern & Floeter, 2008*).

## Statistical analyses

All statistical analyses were done using site means as replicates because chronic anchor damage is more appropriately assessed at the site-level than the transect-level, and because sites are a meaningful unit for management. We tested effects of two categorical factors: anchoring activity (a fixed factor with 3 levels: high (H), medium (M), and low (L)), and group (a random effect with 10 levels). Although sea fans were included in all surveys for benthic cover, specific measurements of sea fan density and size were made at fewer sites ($n = 15$). These 15 sites were classified as high ($n_H = 8$) or low ($n_L = 7$) anchoring.

When data conformed to the assumptions of analysis of variance (ANOVA), we used randomized block ANOVAs to assess the impact of anchoring while accounting for effects of geographic proximity (group), and pairwise comparisons between anchoring activity levels (H-L, H-M, L-M) were made using least squares means (LSM). When data did not conform to the ANOVA assumption of normality, we were sometimes able to transform the data so that the assumptions were met. In other cases, we used either nonparametric Kruskal-Wallis (K-W) tests with Nemenyi post-hoc tests or Maximum Likelihood Estimation (MLE) models using appropriate distributions to model the variance. For MLE models, changes in goodness-of-fit statistics are typically used to evaluate the contribution of subsets of explanatory variables to a particular model, so MLE models were fitted with anchoring only, group only, and both anchoring and group as factors. These models were then compared using Akaike Information Criterion (AIC) and the parameters from the best fitting model were used to calculate means and standard errors for an intuitive presentation of results (*Bolker, 2008*). MLE models for proportion data utilized a beta distribution with parameters 'a' and 'b,' from which we calculated the mean (a/(a+b)) and the variance ($ab/((a+b)^2(a+b+1))$). MLE models for data that were positive and continuous utilized gamma and lognormal distributions. The gamma distribution resulted in 's' (scale) and 'a' (shape) parameters, with which we calculated means (as) and variances ($as^2$). The lognormal distribution provided $\mu$ and $\sigma$ parameters, with which we calculated means ($\exp(\mu + \sigma^2/2)$) and variances ($\exp(2\mu + \sigma^2)(\exp(\sigma^2) - 1)$). Standard error was calculated as the square root of the variance. We report details of test results in a (Table S1) and list only relevant *p*-values for pairwise comparisons between levels of anchoring activity in the text. For ease of interpretation and comparison between anchoring intensities, we have reported parameter estimates, *z*-values, and *p*-values only from models comparing across anchoring levels. All analyses were conducted in R v. 3.0.2 (*R Core Team, 2013*), using packages 'bbmle', 'lsmeans' and 'PMCMR'.

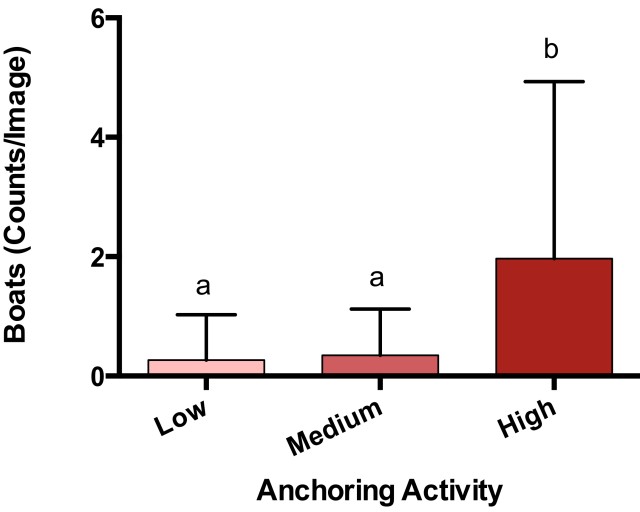

**Figure 2  Anchoring activity observed using satellite imagery.** Sites were classified as high, medium or low anchoring based on expert opinion. This classification was checked by counting anchored boats at each site. Plotted are means (±SE) and letters above bars indicate significant differences based on a multiple comparison test (means that do not differ share a letter). Map Data from GoogleEarth and Digital-Globe.

## RESULTS

### Checking the classification of anchoring activity and confounding with land-based impacts

The observed density of anchored yachts and frequency of anchor damage symptoms corroborated the expert-based classification of anchoring activity. Based on surveys of satellite imagery, there were more boats anchored at sites classified as high anchoring than either medium or low anchoring ($p < 0.0001$) (Fig. 2). On the reef itself, high anchoring sites also showed the most signs of apparent anchor damage. We encountered more overturned scleractinian corals ($p_{H-L} = 0.046$), broken scleractinian corals ($p_{H-L} = 0.048$), and broken gorgonians ($p_{H-L} = 0.002$) at high-anchoring sites, and these symptoms were substantially reduced at low-anchoring sites (Fig. S3). Overall, the post-hoc checks corroborated the distinction between low- and high-anchoring sites, and confirmed that, although the 3 medium-anchoring sites show intermediate levels of damage symptoms (Fig. S3), they cannot be conclusively separated from high- and low-anchoring sites.

We also confirmed that there were no apparent differences in proximity to land-based stressors. The low-, medium-, and high-anchoring sites were at similar distances from both land ($p = 0.60$) and development ($p = 0.75$) (Fig. S4).

### Responses of benthic taxa associated with anchoring

Of the taxa whose benthic cover was surveyed, only hard corals and sea fans showed a significant response to anchoring activity (Table 1). Hard coral cover at highly anchored sites was reduced by a factor of roughly 1.7 relative to sites experiencing medium or little anchoring ($p_{M-H} = 0.047$, $p_{L-H} = 0.02$, $p_{L-M} = 0.89$) (Fig. 3). Sea fan cover at sites

**Table 1  Percent cover of major benthic taxa at sites with different levels of anchoring.** Mean cover (± 95% CI) at sites with low (L), medium (H) and high (H) levels of anchoring, and the proportional change between high and low levels. The last column indicates which groups are significantly different from each other using a multiple comparison test (see Methods for further details).

| Taxonomic group | Low anchoring sites (% cover) | Medium anchoring sites (% cover) | High anchoring sites (% cover) | Proportional change in cover (H-L) /L | Sites that differed |
|---|---|---|---|---|---|
| Hard Coral | 17.05 ± 2.4 | 16.06 ± 2.3 | 9.91 ± 1.9 | −0.42 | H-L, H-M |
| Fire Coral | 3.65 ± 0.5 | 3.09 ± 0.9 | 2.73 ± 0.5 | −0.25 | None |
| Sea Fan | 12.78 ± 5.9 | 4.71 ± 3.9 | 5.00 ± 4.0 | −0.61 | H-L, M-L |
| Soft Branching Coral | 18.58 ± 6.3 | 19.15 ± 6.4 | 13.20 ± 5.7 | −0.29 | None |
| Sponges: Erect | 3.77 ± 2.8 | 5.93 ± 4.7 | 1.86 ± 0.9 | −0.51 | None |
| Sponges: Encrusting | 5.07 ± 1.5 | 3.77 ± 2.7 | 2.93 ± 1.6 | −0.42 | None |
| Algae: Fleshy | 12.85 ± 5.8 | 14.79 ± 6.1 | 20.74 ± 6.7 | 0.61 | None |
| Algae: Calcareous | 2.15 ± 4.6 | 0.70 ± 1.0 | 2.61 ± 3.7 | 0.21 | None |
| Algae: Crustose | 8.69 ± 6.5 | 6.95 ± 2.2 | 7.47 ± 3.8 | −0.14 | None |
| Algae: Filamentous | 1.25 ± 0.7 | 2.65 ± 3.6 | 2.33 ± 2.0 | 0.86 | None |
| Dead Coral | 2.57 ± 2.0 | 2.80 ± 3.7 | 6.47 ± 3.55 | 1.52 | None |

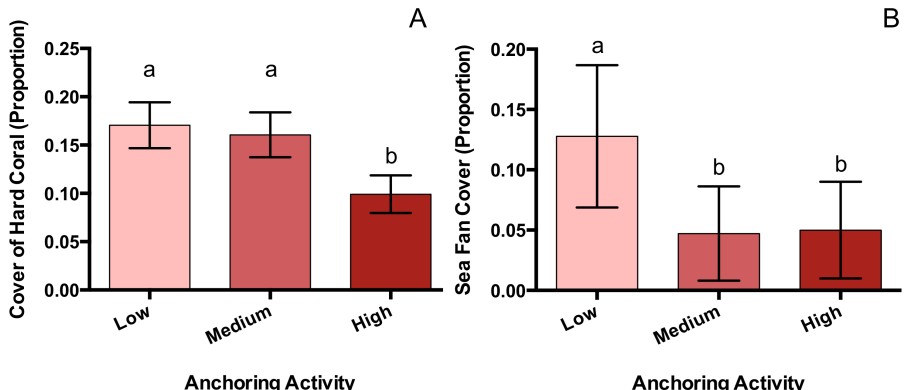

**Figure 3  Hard coral and sea fan cover differ across a gradient of anchoring activity.** Means (±SE) of proportional bottom cover for scleractinian corals (A) and sea fans (B). For each taxon, letters above bars indicate significant differences based on a multiple comparison test (means that do not differ share a letter).

with high and medium levels of anchoring was reduced by a factor of 2.6 compared to low-anchoring sites ($p_{M-H} = 0.9$, $p_{H-L} = 0.044$, $p_{L-M} = 0.049$) (Fig. 3).

## Responses of coral populations associated with anchoring

Coral colonies were approximately 39% smaller at sites with high levels of anchoring activity than at sites with medium or low levels ($p_{L-H} = 0.046$, $p_{M-H} = 0.28$, $p_{L-M} = 0.996$) (Fig. 4). In addition, coral colony density was roughly 57% lower at sites with high anchoring than at sites with little or no anchoring ($p_{L-H} = 0.0003$, $p_{M-H} = 0.55$, $p_{L-M} = 0.068$) (Fig. 4). Coral species richness was also reduced by approximately 42% at sites with higher anchoring activity (either medium or high) than those with low levels ($p_{H-L} = 0.0002$, $p_{H-M} = 0.94$, $p_{L-M} = 0.01$) (Fig. 4).

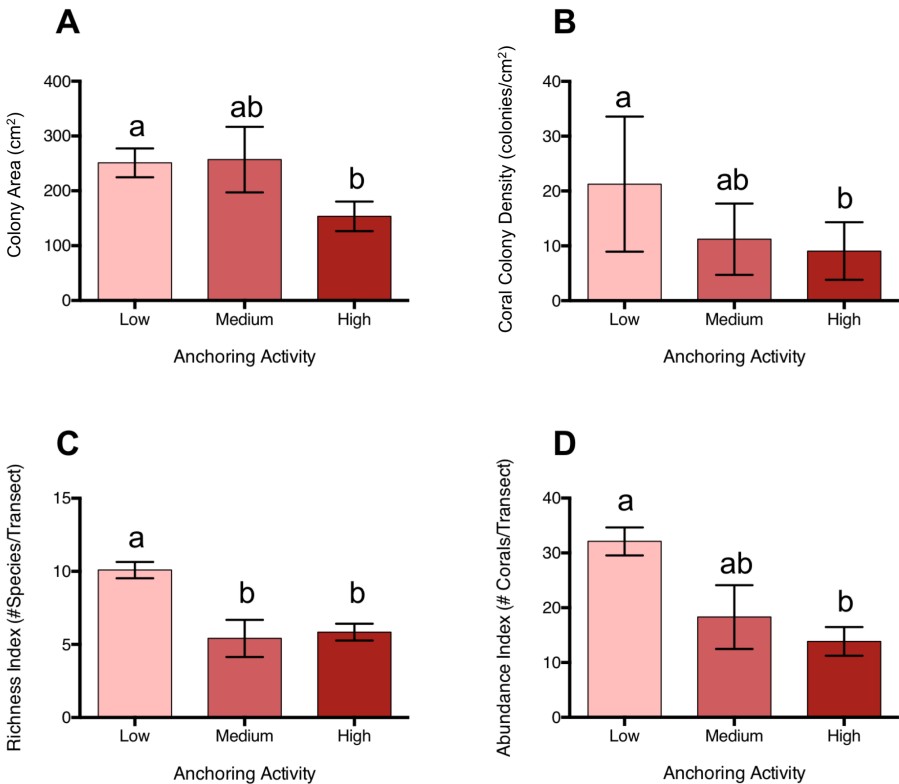

**Figure 4 Coral diversity, abundance and size differ across a gradient of anchoring activity.** Plotted are means (±SE) of (A) Coral colony surface area, (B) coral colony density, (C) coral species richness and (D) coral abundance. Letters above bars indicate significant differences based on a multiple comparison test (means that do not differ share a letter).

Corals with branching, mounding and plate morphologies were most strongly affected by anchoring (Table S2). Branching corals included *Acropora cervicornis, Madracis decactis, Porites divaricata, P. furcata,* and *P. porites*; mounding corals were primarily *Dichocoenia stokesii, Favia fragum, Madracis pharensis, Montastraea cavernosa, Orbicella annularis, O. faveolata, O. franksii, P. astreoides, Siderastrea radians*, and *S. siderea*, and plate corals were *Agaricia agaricites, A. humilis*, and *A. lamarcki*. Branching corals were roughly 65% smaller in size ($p = 0.05$) and their colony density was reduced by approximately 67% at high anchoring sites than at low-anchoring sites ($p_{H-L} = 0.041$, $p_{H-M} = 0.97$, $p_{L-M} = 0.37$). Similarly, both the colony surface area ($p_{H-L} < 0.0001$, $p_{H-M} = 0.03$, $p_{L-M} = 0.65$) and colony densities ($p_{L-H} = 0.002$, $p_{M-H} = 0.69$;, $p_{M-L} = 0.10$) of mounding corals were reduced by roughly half at sites with high anchoring. Plate coral surface area did not differ between anchoring intensities (Table S2), but colony density was 55% lower at high anchoring sites ($p_{L-H} = 0.026$, $p_{M-H} = 0.61$, $p_{M-L} = 0.03$). Brain corals, cup-like corals, and encrusting corals did not differ in surface area or colony density across anchoring regimes (Table S2).

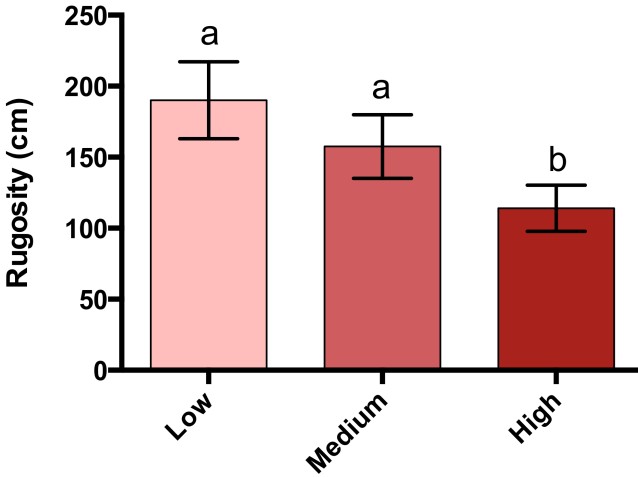

**Figure 5  Rugosity differs across a gradient of anchoring activity.** Plotted are means (±SE) of reef rugosity and letters above bars indicate significant differences based on a multiple comparison test (means that do not differ share a letter).

## Changes in reef rugosity associated with anchoring

Rugosity at sites with high anchoring activity was 40% lower than at sites with little or no anchoring ($p_{L-H} = 0.00003$, $p_{M-H} = 0.009$, $p_{M-L} = 0.63$) (Fig. 5).

## Changes in fish density associated with anchoring

Mean total fish density was 55% lower at sites with high anchoring than at sites with little or no anchoring ($p_{H-L} = 0.005$), and sites with medium anchoring activity had intermediate densities ($p_{H-M} = 0.4$, $p_{L-M} = 0.48$) (Fig. 6). This reduction was driven by adult fish, whose density was 66% lower at low-anchoring sites than at high-anchoring sites ($p_{H-L} = 0.002$), with intermediate densities at medium-anchoring sites ($p_{H-M} = 0.2$, $p_{L-M} = 0.58$). The density of juvenile fish did not differ by anchoring level ($p_{H-M} = 0.2$, $p_{H-L} = 0.6$, $p_{L-M} = 0.08$). Accompanying the decline in total fish density was a corresponding decline in fish species richness, which was 35% lower at highly anchored sites compared to sites with little or no anchoring ($p_{H-L} = 0.0004$); and richness was intermediate at sites with medium-levels of anchoring ($p_{H-M} = 0.6$, $p_{L-M} = 0.06$) (Fig. 6).

These reductions in fish density were spread across several trophically based functional groups (Table 2). Herbivores capable of removing substratum while feeding (adult scrapers and excavators), were reduced in density by roughly 73% at high-anchoring sites relative to medium- and low-anchoring sites ($p_{H-L} = 0.006$, $p_{H-M} = 0.4$, $p_{L-M} = 0.5$). At our sites, excavators were primarily stoplight parrotfish (*Sparisoma viride*), and scrapers were primarily blue parrotfish (*Scarus coeruleus*) and queen parrotfish (*Scarus vetula*). Other types of adult herbivore, who vary in feeding behavior but do not excavate the substratum, were 66% lower in density at high-anchoring sites than elsewhere ($p_{H-L} = 0.002$, $p_{H-M} = 0.14$, $p_{L-M} = 0.7$). These other herbivores were primarily other

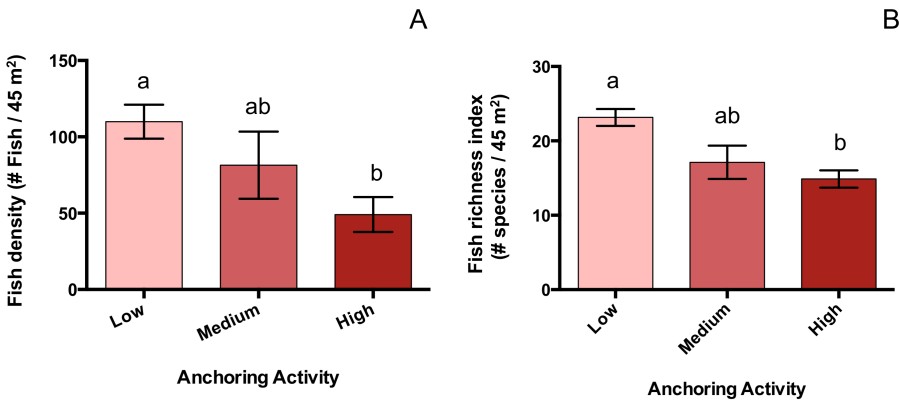

**Figure 6 Fish abundance and diversity differ across a gradient of anchoring activity.** Plotted are means (±SE) of fish density (left) and species richness (right). Letters above bars indicate significant differences based on a multiple comparison test (means that do not differ share a letter).

**Table 2 Densities of major fish functional groups at sites with different levels of anchoring.** Mean fish density ± 95% CI at sites with low (L), medium (H) and high (H) levels of anchoring, and the proportional change between high and low levels. The last column indicates which groups are significantly different from each other using a multiple comparison test (see Methods for further details).

| Fish functional group | Low anchoring sites (#/45 m$^2$) | Medium anchoring sites (#/45 m$^2$) | High anchoring sites (#/45 m$^2$) | Proportional change in density (H-L) /L | Sites that differed |
|---|---|---|---|---|---|
| Scrapers and excavators | 2.84 ± 0.39 | 1.94 ± 0.77 | 0.75 ± 0.4 | −0.73 | H-L |
| Other herbivores | 43.25 ± 4.72 | 34.92 ± 9.34 | 13.69 ± 4.86 | −0.68 | H-L |
| Piscivores | 3.11 ± 2.21 | 2.73 ± 1.94 | 1.21 ± 0.86 | −0.61 | H-L, H-M |
| Spongivores | 0.457 ± 0.38 | 0.23 ± 0.32 | 0.025 ± 0.08 | −0.94 | H-L |
| Benthic Carnivores | 12.86 ± 6.17 | 14.63 ± 7.02 | 8.26 ± 3.97 | −0.35 | H-L, H-M |
| Planktivores | 19.85 ± 22.42 | 12.86 ± 10.1 | 6.86 ± 6.59 | −0.65 | None |
| Omnivores | 1.42 ± 0.73 | 1.96 ± 1.18 | 1.62 ± 1.66 | 0.14 | None |

parrotfish, surgeonfish, and damselfish and were functionally turf grazers, macroalgal browsers and territorial algal/detritus feeders.

Other functional groups of fish that consume benthic resources also displayed reductions in density associated with anchoring (Table 2). Spongivore densities were 95% lower at highly anchored sites than at low-and medium-anchoring sites ($p_{H-L} = 0.003$, $p_{H-M} = 0.4$, $p_{M-L} = 0.4$) and comprised primarily angelfish, filefish, spadefish, and boxfishes. Benthic carnivores were reduced by 36% at high-anchoring sites ($p_{L-H} = 0.03$, $p_{M-H} = 0.03$, $p_{M-L} = 0.63$). Benthic carnivores at our sites included species that consume mobile invertebrates or sessile invertebrates, and some may also eat fish. This was a broad group that included wrasses, butterflyfishes, hamlets, some groupers, bass, pufferfish, grunts, goatfish, drums, snappers, trunkfish, triggerfish, mojarra, porgies, and porcupine fish. Lastly, there were also 61% fewer piscivores at high-anchoring sites than at low- and medium-anchoring sites ($p_{M-H} = 0.012$, $p_{L-H} = 0.004$, $p_{M-L} = 0.73$). At our BVI sites, piscivores comprised mostly groupers, lizardfish and lionfish.

### Fraction of reefs vulnerable to anchoring

From the marine habitat maps we examined, we estimated 24% of coral reef area in the British Virgin Islands is leeward in exposure, although this only a rough estimate of the fraction of reef that is safe to anchor near because exposure varies with the seasons and weather.

## DISCUSSION

Although we attempted to control for other factors that might impact reefs, it is difficult to reliably decouple the effects of anchoring from other human impacts. Increased boat activity is associated with more anchor damage (Fig. S3), but there may be other factors coupled to human visitation affecting reef communities in these locations. We argue that damage by snorkelers and divers from the boats is unlikely to be a major source of confounding because (1) most sites were too deep for recreational snorkelers to plausibly damage the reef, (2) most recreational diving is concentrated at other sites at which the local dive operators use moorings specifically designated for dive boats, and (3) damage symptoms commonly included smashed or overturned corals that appear too large (> 0.5 m diameter) to be attributed to divers (Fig. S3). Nonetheless, some of the impacts we attribute to anchoring could also be caused by sewage discharge from boats, boat hull antifouling paint, and littering by boat users. In addition, this study presents a snapshot in time. It is possible that boating impacts have already interacted with other stressors at the site and may have diminished anchored reefs' recovery from other stressors (*Carilli et al., 2009*).

While anchoring impacts have been documented for soft sediments (*Backhurst & Cole, 2000*), seagrass beds (*Francour, Ganteaume & Poulain, 1999*; *Creed & Amado Filho, 1999*; *Milazzo et al., 2004*) and associated taxa (*Hendriks et al., 2013*), we are aware of few previous studies documenting community-wide impacts associated with anchoring on coral reefs. Previous work has documented the mechanisms by which anchors damage reefs (*Glynn, 1994*; *McManus, Reyes & Nañola, 1997*; *Dinsdale & Harriott, 2004*; *Fava et al., 2009*), and used models to predict rates of coral-cover loss (*McManus, Reyes & Nañola, 1997*; *Saphier & Hoffmann, 2005*). Early empirical studies imply that impacts can be substantial. In Florida, extensive (20% by area) anchor damage to *A. cervicornis* thickets was observed (*Davis, 1977*), and circumstantial evidence linked anchoring to the decline of *A. palmata* and other corals in this area (*Dustan & Halas, 1987*). In the Pacific, Edinger and co-workers (*1998*) argued that multiple human activities, including anchoring, affected Indonesian coastal reefs, but perhaps the strongest evidence for anchoring impacts comes from the Great Barrier Reef where, at three sites subject to frequent anchoring, symptoms of coral damage were increased, and the cover of three coral families and of soft corals was reduced relative to three equivalent low-anchoring sites (*Dinsdale & Harriott, 2004*).

We isolated a surprisingly substantial reduction in coral cover associated with boat anchoring in the BVI. Our study was not designed to measure the effect of anchoring against other human and natural agents of change but, for context, we note that mean coral cover at our low-anchoring sites (17.1%) is above the recent Caribbean-wide average

of 14.3%, whereas cover at the high anchoring sites (9.9%) is just above the region-wide 25% quantile (*Jackson et al., 2014*). The difference in absolute coral cover between high- and low-anchoring sites (7.2%) is also substantial relative to long-term declines in coral cover measured in the BVI (12% over 21 years) and region-wide (19% over 40 years) (*Jackson et al., 2014*; *Forrester et al., 2015a*). In a related BVI study, there was no obvious recovery after 11 years from the damage caused by a one-off anchoring event (*Forrester et al., 2015b*), suggesting that the spatial gradients we measured are likely to persist unless anchoring activity declines.

The community-wide impacts associated with anchoring appeared to comprise a mix of direct and indirect effects. Abundant symptoms of damage to coral colonies at high-anchoring sites, plus the fact that impacts were most severe for corals with high vertical relief (branching, plating and mounding colonies), suggests damage from anchor chains sweeping across the reef is a major direct agent of coral loss (*Davis, 1977*; *Forrester et al., 2015b*). The same reasoning suggests that mortality from contact with anchors and chains accounts for most of the loss of sea fans at high-anchoring sites. These taxa contribute much of the vertical relief on coral reefs, so the approximate halving of their abundance explains why reef rugosity was reduced by almost half at frequently anchored sites. Because many reef fish taxa depend on three-dimensional reef structure for shelter (*Luckhurst & Luckhurst, 1978*; *Ault & Johnson, 1998*), the 'flattening' of highly anchored sites is a plausible explanation for reduced densities of most (5 of 7) functional groups of adult fishes. Juvenile fish densities did not differ across anchoring intensities, suggesting that juveniles may be generally less-sensitive to habitat damage associated with anchoring than adults. The lack of detectable anchoring effects on some benthic taxa (e.g., encrusting sponges, fire coral, crustose coralline and filamentous algae) may be due to their low profile and lower vulnerability to anchor damage, or to rapid recovery rates after damage. Reefs that have experienced reductions in coral cover and in the abundance of herbivorous fish commonly show elevated biomass of fleshy macroalgae (*Adam et al., 2015*). In the BVI, although mean cover of fleshy macroalgae almost doubled at high anchoring sites, this 'trend' was not significant suggesting that further sampling is needed to resolve the macroalgal response to anchoring.

Impacts associated with boat anchoring, relative to other human impacts, remain hard to extrapolate. We estimated that roughly 24% of BVI reefs were suitable for anchoring under most conditions. A more sophisticated vulnerability assessment indicated that 19% of the Great Barrier Reef World Heritage Site is vulnerable to anchor damage, primarily in areas where coral reefs or seagrass beds coincide with likely anchor deployment (*Kininmonth et al., 2014*). Because anchor damage appears to be a substantive contributor to coral reef decline in the BVI, but its effects elsewhere are poorly documented, we argue that further assessments of actual anchoring impacts in other areas are worthwhile.

The magnitude of differences between sites with and without chronic anchoring was surprising given the BVI's network of mooring buoys that dates to the 1970s (L Jarecki, Guana Science, BVI, pers. comm., 2012). Currently, there are 66 sites with ~200 moorings managed by the National Parks Trust in the BVI (N Pascoe, National Parks Trust of the Virgin Islands, pers. comm., 2014), plus additional "unofficial" mooring sites. Our informal

observations at high-anchoring sites with moorings revealed boats anchored on reef, often when all moorings were in use, but sometimes even when moorings were available. In other instances, boats anchored on sand in and around moorings, but their anchor chains draped across adjacent reef and swept back and forth across the reef's surface as shifting tides and winds swung the boats on their anchors. Boaters weigh multiple factors when deciding whether and where to anchor (*Gray et al., 2010*; *McAuliffe et al., 2014*), including potential damage to benthic habitats (*Diedrich et al., 2013*). In principle, therefore, anchor damage can be mitigated by effective local education and management programs and represents a more tractable problem than many global stressors, such as ocean warming (*Beeden et al., 2014*).

## CONCLUSIONS

The impacts associated with chronic boat anchoring in the BVI were of surprisingly high magnitude, and were linked to a roughly 50% reduction in coral and sea fan abundance. Indirect impacts on reef rugosity and on many common families of fish were of broadly similar magnitude. The magnitude of these impacts was surprising, in part, because of the well-developed network of mooring buoys in the area designed to reduce the frequency of anchoring. Because the magnitude of impact was estimated based on a spatial comparison, it is hard to directly compare the effects of anchoring to those of other anthropogenic stressors whose impacts are more commonly assessed by temporal before-after comparisons. Nonetheless, we suggest that further analysis of boat anchoring effects is worthwhile and that the priorities should be (1) estimating the magnitude of boating impacts relative to other agents of reef damage and (2) understanding how moorings can be best used to mitigate anchor damage.

## ACKNOWLEDGEMENTS

We are grateful to Jessica Perreault, David Gleeson, and Kristian Dzilenski for field assistance. Earlier versions of the manuscript were improved by input from Lianna Jarecki, Jason Kolbe, Gavino Puggioni, Mary Donovan, Peggy Fong and an anonymous reviewer. Finally, thanks to the local experts in the BVI who shared their knowledge with us.

### Funding

This work was funded by The Nature Conservancy's Global Marine Team and The Falconwood Foundation, and the Guana Island Staff, Dive BVI, and UBS Divers provided logistical support. The funders had no role in study design, data collection and analysis, decision to publish, or preparation of the manuscript.

### Grant Disclosures

The following grant information was disclosed by the authors:
The Nature Conservancy's Global Marine Team.

Falconwood Foundation.
Guana Island Staff.
Dive BVI.
UBS Divers.

## Competing Interests

The authors declare there are no competing interests.

## Author Contributions

- Rebecca L. Flynn and Graham E. Forrester conceived and designed the experiments, performed the experiments, analyzed the data, prepared figures and/or tables, authored or reviewed drafts of the paper, approved the final draft.

## Data Availability

The raw data are available as Supplemental Files.

## Supplemental Information

Supplemental information for this article can be found online at http://dx.doi.org/10.7717/peerj.7010#supplemental-information.

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
