# Peer review of "Boat anchoring contributes substantially to coral reef degradation in the British Virgin Islands"

_PeerJ, doi:10.7717/peerj.7010_

## Round 0.1 · original submission · Major Revisions

I now have three reviews of your manuscript from peers who are familiar with your research area. All of them like the contribution in general, and some of the comments will only require minor revision. The important criticism is highlighted by Reviewer 1 and noted by Reviewer 3, and concerns the seemingly qualitative nature of site classification -- this needs to be addressed by clarifying the classification or by addressing this weakness in the method clearly in the discussion.

Please include a point-by-point response to all of the reviewers' comments with your revised ms., which I will send back for further review.

Reviewer 1 ·

Basic reporting

no comment

Experimental design

My primary concern relates to the classification of sites into high, medium, and low anchoring categories. While I was initially impressed by the use of a combination of local experts and satellite images, there were inconsistencies in the data from these two sources. For example, Long point and Monkey point are categorized as “medium” (per the recommendation of local experts), but zero boats were observed in satellite images, implying an absence of anchoring pressure at these sites. Moreover, eight boats were counted in satellite images of “low” anchoring Lee Bay, and two boats were counted in satellite images of “high” anchoring Crab Cove. These inconsistencies suggest that there is not enough justification for how sites were classified, and this needs to be improved. Furthermore, it was concerning that the values given for the number of sites in each category -among the text (Line 72-73), figures (Fig 1), and data (AnchHistGE.csv)- did not match. Please clarify how many sites were in each category, perhaps with a table listing each site, and which paired “group” and anchoring category it fell into. Finally, low and medium anchoring sites look to be statistically indistinguishable (Fig 2). What was the reasoning for distinguishing between them?

Additional Comments:

Line 86: What point was used to determine distance to shore? Each site is an embayment, which by definition is surrounded by shore. Was the reef used as the origin point? If so, how far was each reef to where boats were actually anchored?

Lines 118-120: Specify how fish were categorized into feeding guilds and juvenile/adult.

Lines 122-148: The description of statistical analyses was very confusing; please re-word for general clarity, and specify which tests were used for each variable.

Validity of the findings

The discussion contained a nice synthesis of the findings with existing literature, but there was not enough mention of the limitations of this study. Though some factors (such as depth, distance from shore), were controlled for, it is difficult to decouple the effects of anchoring from other human impacts. More boats mean more anchor damage, but also perhaps more fin-strikes by snorkelers and scuba divers, sewage/littering, and chemicals leaching from boat-bottoms. These factors have all been implicated in local-scale harm of reef-associated biota (Donohue et al. 2001; Dearden et al. 2010; Burgin and Hardiman 2011[cited in ms]; Titley-O’Neal et al. 2011; Webler and Jakubowski 2016; Renfro and Chadwick 2017). Similarly, it is too speculative to say that chronic boat anchoring “caused” the observed reductions in coral and sea fan abundance (to paraphrase lines 332-333); all declarations of causation should be replaced with language that is more representative of the tests that were conducted.

Additional Comments:

Lines 300-302: There also may be other factors controlling juvenile fish density, such as the presence of nearby seagrass, even at high anchoring sites.

Lines 302-303: There is no mention of these groups in the results.

References:
Dearden P, Theberge M, Yasue M (2010) Using underwater cameras to assess the effects of snorkeler and SCUBA diver presence on coral reef fish abundance, family richness, and species composition. Environ Monit Assess 163:531–538. doi: 10.1007/s10661-009-0855-3

Donohue MJ, Boland RC, Sramek CM, Antonelis GA (2001) Derelict fishing gear in the Northwestern Hawaiian Islands: Diving surveys and debris removal in 1999 confirm threat to coral reef ecosystems. Mar Pollut Bull 42:1301–1312. doi: 10.1016/S0025-326X(01)00139-4

Renfro B, Chadwick NE (2017) Benthic community structure on coral reefs exposed to intensive recreational snorkeling. PLoS One 12:e0184175. doi: 10.1371/journal.pone.0184175

Titley-O’Neal CP, MacDonald BA, Pelletier E, Saint-Louis R, Phillip OS (2011) The Relationship Between Imposex and Tributyltin (TBT) Concentration in Strombus Gigas from the British Virgin Islands. Bull Mar Sci 87:421–435. doi: 10.5343/bms.2010.1093

Webler T, Jakubowski K (2016) Mitigating damaging behaviors of snorkelers to coral reefs in Puerto Rico through a pre-trip media-based intervention. Biol Conserv 197:223–228. doi: 10.1016/j.biocon.2016.03.012

Additional comments

My general response to this manuscript was positive, and I especially appreciated the inclusion of less traditionally popular groups like soft branching corals, sea fans, and sponges in the surveys. With careful revision, this manuscript could be reconsidered for additional review. Below I offer some additional general comments:

Lines 149-260: The amount of parenthetical text/data made this section very difficult to read. The parenthetical information pertaining to statistical tests, which sometimes occupied up to four lines (Lines 191-194), could be moved to a table.

Lines 155-161: Consider moving these sentences to the section describing rugosity (Lines 220-224).

Lines 181-218: This gets redundant, especially because differences in surface area and density are the same, regardless of whether corals are grouped by morphology or taxonomically by genus, and they also reflect the total coral results as a whole.

Line 184: The accepted classification is Montastrea cavernosa (WoRMS 2019).

Fig 1: Please consider labelling the major islands (e.g., Tortola, Virgin Gorda). It may also be helpful to add abbreviated site labels, or markers distinguishing paired groups.

Fig 2: Please remove the caption information about multiple comparison letters, and where map data are from, since there are no letters or maps in this figure.

Fig 5: Please change “proportional bottom cover” to “rugosity” in the figure caption.

References:

WoRMS (2019). World Register of Marine Species. Available from: http://www.marinespecies.org/aphia.php?p=taxdetails&id=287962

·

Basic reporting

This is a very well-written and straight forward manuscript that assesses the damage anchoring boats have caused to coral reefs of the British Virgin Islands. It reports the results of an extensive field survey of 25 reef sites, quantifying benthic community structure, physical reef structures, and associated ecosystem functions such as providing fish habitat. They found clear and unambiguous evidence of substantial damage across all measures.

Experimental design

This Ms clearly falls within the scope of the journal and fill a well-defined knowledge gap. Truly, when I first read this, I was doubtful that this question had not already been addressed fully. I know that boat moorings have been placed in popular anchorages throughout Biscayne and US Virgin Islands National Parks, and I assumed that was in response to studies of anchor damage. I also know that the US Park Service in the USVI has long-term survey data around many of their moorings on St John as I have seen these data. However, they are all on sandy substrates covered in seagrasses, including an invasive species. To my surprise, the documentation of damage of anchoring to hard bottom habitat appears to be sparse as the authors suggest. This research is rigorous and the methodology complete

Validity of the findings

As stated above, the data are strong and well analyzed. The single criticism I have is that the presentation of the statistical results as parenthetical statements make this Ms really hard to read and follow. I am an ecologist, so I am used to this type of reporting, but even for me, it was too much. For a more genera audience, I fear this is prohibitively distracting. I think these statistics need to be moved to a table, where those that are really interested can study all of the details. I understand that there are a lot of different transformations and analyses due to the nature of the data, so a creating an appropriate table is not trivial. However, I think it is key to make the Ms more digestible to a general audience

Additional comments

I think this Ms is of broad enough scope that it merit publication in PeerJ

·

Basic reporting

The article is well composed and readable. The introduction provides a nicely constructed background for the study, includes relevant literature, and the objectives and hypotheses are clear. The study fills an important knowledge gap about how boat anchoring translates to coral reef benthic cover and structure.

The structure of the article meets the journal’s standards, and the figures are relevant. I have a few suggestions to improve readability:

Results – this section was difficult to read as it is composed more like a list of tests than a narrative of results. Perhaps you can add some of these results as tables and refer to them that way, and use this section to highlight the most important results.

Table 1 – spaces in between the mean and ± would make it easier to read
Figure 1 – need to indicate in some way the location of the map (e.g., include latitude/longitudes).
Figure 3 caption – remove ‘Plotted are’
Figure 4 – fix superscript in y-axis labels
Figure 6 – right hand side of the figure is cut off in the pdf

Also, the data used in the analyses are attached, but could benefit from some data descriptors/metadata. Please describe what each file consists of, what data are in each column, what are the units, etc.

Experimental design

The data were collected in a sound manner, and statistical analyses were appropriate to the data and questions being invested. A potentially minor issue is that I found the classification of low-medium-high boat traffic to be a bit circular. These classifications were made by ‘expert opinion’ and then later backed up by counting boats in satellite images. Why not use the counts to classify the sites? Low/medium sites didn’t have a significant difference in #boats (Fig 1), so perhaps the analyses should lump the low/medium sites? Would that change the conclusions?

Line 306 – reference to Jackson et al here not the best reference as we didn’t explicitly test this. I suggest referencing Adam et al 2015 MEPS:520 for a review of the topic.

Validity of the findings

The authors did a nice job of tempering their conclusions given that they could not assess the effects of boat anchoring in the context of other anthropogenic stressors, and the conclusions were well justified by the results.

Overall the study was well thought out, the manuscript was nicely composed, and I have relatively few and minor comments such that I recommend the publication of this study.

---

## Round 0.2 · accepted · Accept

Please note the comments of Reviewer 1 on the revised manuscript in preparing a final version for publication.

Reviewer 1 ·

Basic reporting

no comment

Experimental design

The authors greatly improved the methods section, and the new table is a good addition. I also appreciate the further justification of sites into their respective anchoring categories, and description of the rationale for choosing a "medium" anchoring category. However, I did notice some incorrectly defined/labeled terms while going through this revised ms. First, the standard error is incorrectly defined in lines 170-171, and this is the definition of the standard deviation. The standard error = standard deviation divided by the square root of the sample size. Similarly, the error bars in the bar graphs show the 95% CI, not the standard error. I recommend correcting the bar graphs so that error bars represent SE, as this is more relevant for comparing groups.

Validity of the findings

The authors have done a nice job of addressing the limitations of their study, and rephrasing some of their previously speculative language.

Additional comments

This revised manuscript is much improved, and I agree with the other reviewers that this work addresses an important knowledge gap, and aligns with the scope of the journal. After the minor revisions mentioned above, I consider this manuscript ready for publication.